molecular biology/immunology

RNA-sequencing, sheep, mastitis, mammary gland, immune response

**Author for correspondence:**
Youji Ma
e-mail: yjma@gsau.edu.cn

# Digital gene expression analyses of mammary glands from meat ewes naturally infected with clinical mastitis

Taotao Li[1], Jianfeng Gao[1], Xingxu Zhao[2] and Youji Ma[1]

[1]College of Animal Science and Technology, and [2]College of Veterinary Medicine, Gansu Agricultural University, Lanzhou, People's Republic of China

TL, 0000-0003-4391-3291; YM, 0000-0002-4328-5654

Clinical mastitis in sheep has gravely restrained production performance for a long time. Knowledge of mechanisms of its pathogenesis and resistance in meat sheep mammary gland with clinical mastitis are not yet understood, especially for clinical mastitis caused by natural infection. In this work, RNA-sequencing was firstly used to screen the differentially expressed genes (DEGs) in clinical mastitic mammary tissues (CMMTs) when compared with healthy mammary tissues (HMTs) from meat sheep flocks. We identified 420 DEGs including 316 upregulated and 104 downregulated genes in CMMTs. Gene ontology annotation revealed these DEGs were mainly engaged in immune response and inflammation response. Pathway enrichment showed they were primarily enriched in pathways relevant to inflammation, immune response and metabolism. Alternative splicing analysis showed most common differential splicing genes in CMMTs and HMTs were implicated in immune response. Immunostaining for three immune response-related proteins encoded by DEGs were mainly observed in mammary epithelium from both CMMTs and HMTs, and their positive signals were more intensive in CMMTs than those in HMTs. These findings provide experimental basis and reference for further researching the molecular genetic mechanisms, particularly immune defence mechanisms, of sheep mammary gland during clinical mastitis.

## 1. Introduction

Mastitis is a very common disease in mammals, especially cattle and sheep. Mastitis is inflammation of the mammary gland caused by various causative organisms (e.g. bacterium, viruses and mycoplasma) [1], mechanical trauma and physiological or metabolic changes [2]. It is hard to eradicate and it causes

immeasurable economic loss for the developing cattle or sheep production. For sheep flocks, mastitis leads to an abatement and decrease in milk yield and quality (milk appearance and composition) [1,3], thereby affecting the normal growth, development and survival of suckling lambs. Furthermore, mastitis significantly raises treatment cost, and ewes may die in severe cases of mastitis [3]. In animal welfare, sheep with clinical mastitis are characterized by anxiety, restlessness, feeding behaviour changes and pain [1]. Additionally, many studies have reported that clinical mastitis in cows negatively affects the reproductive performance, i.e. oestrous cycle and follicular development [4], conception rate [5], pregnancy rate [5,6], days to first insemination [7], etc. The incidence of clinical mastitis in sheep is generally less than 5% [8], but reached up to 9.4–17.4% in Gavojdian et al.'s [9] reports. There was a close correlation between the occurrence of ovine clinical mastitis and breed, parity and litter size [10]. Until now, most research on molecular mechanisms of mastitis in ruminants has been focused on cattle and is relatively scarce in sheep. Hu sheep (Ovis aries), an unique Chinese indigenous meat sheep breed, is recognized for its early sexual maturity, perennial oestrus and high fertility [11]. It is an ideal resource for researching the molecular pathogenesis of ovine clinical mastitis.

RNA-sequencing (RNA-seq) is now the most widely used method to characterize transcripts, analyse gene expression and uncover the biogenesis and metabolism of RNA, which provides powerful tools for revealing molecular mechanisms in the development, differentiation and disease [12–14]. In recent years, numerous studies on transcriptome analysis of mastitis based on RNA-seq have been conducted on cattle [15–17]. Sheep has been considered an important economic animal, but previous studies on mastitis are mostly centralized in the aetiology [3], epizootiology [18,19], diagnosis [1], control and treatment [1,20] of the disease.

After intra-mammary infection, pathological changes occurring in mammary glands are very intricate, while the molecular mechanisms involved in those changes are still obscure. Currently, the research on transcriptome of mammary glands with mastitis in ewes and its molecular pathogenesis based on RNA-seq worldwide is limited, and relevant studies stay within mastitis caused by experimental bacterial infection [21,22]. Research on differential expression and regulation of the genes in ovine mammary glands naturally infected with mastitis is rarely reported. Thus, this work was designed for the first time to obtain transcriptome and differentially expressed genes (DEGs) in ovine clinical mastitic mammary tissues (CMMTs) when compared with healthy mammary tissues (HMTs) using RNA-seq. It also analyses the function, categories and pathways of DEGs via bioinformatics, and preliminarily screens potential genes associated with mastitis by molecular biology technology, which was necessary to investigate mechanisms during intra-mammary infections.

# 2. Material and Methods

## 2.1. Experimental animals and design

Six multiparous purebred Hu sheep (two years old), half with clinical mastitis and half uninfected as the control group, were provided by the Pingchang Hu Sheep Breeding Base (Lintao, Gansu, China). A clinical examination was performed to ascertain whether the experimental animals were healthy or mastitic. Clinical mastitis is characterized by abnormalities in the udder such as swelling, redness, pain and hardness. Clinical mastitis for all three ewes used in this study was caused by mixed infections with Staphylococcus aureus and Escherichia coli according to Gram staining and biochemical identification. Samples of mammary tissues were collected from these animals. Duplicated samples were collected for all tissues: one sample was immediately placed in liquid nitrogen and subsequently cryopreserved at −80°C for the extraction of total RNA and protein, and the other was fixed with 4% paraformaldehyde for approximately 48 h, dehydrated in a gradient series of ethanol, cleared in xylene and embedded in paraffin. The tissue blocks were sectioned at 5 μm thickness and used for haematoxylin and eosin (H&E) and immunohistochemistry staining.

## 2.2. Total RNA extraction, cDNA library preparation and RNA-seq

Total RNA was extracted from each sample using Trizol Reagent (TransGen Biotech, China) according to instructions. The concentrations and quality of RNA samples were examined by NanoDrop2000 (Thermo Scientific, Waltham, MA, USA) and Agilent 2100 (Agilent, Santa Clara, CA, USA). The cDNA libraries were constructed from 1.5 μg of total RNA per sample using NEBNext® Ultra™ RNA Library Pre Kit (New England Biolabs, Inc.). In brief, mRNA isolation and fragmentation were conducted using

NEBNext® Poly(A) mRNA Magnetic Isolation Module (New England Biolabs, Inc.) and fragmentation buffer, respectively. We used reverse transcriptase and random hexamer-primer to synthesize first-strand cDNA with cleaved short RNA fragments as template. We subsequently synthesized second-strand cDNA by adding reaction buffer, dNTPs, RNase H and DNA polymerase I using first-strand cDNA as template. The double-stranded cDNA was ultimately purified by end repair, adaptor ligationgel, selection and library enrichment successively. The quality assessment and quantification of the sample libraries were performed using Qubit® Fluorometers (Thermo Fisher Scientific) and KAPA Library Quantification Kit (KAPA Biosystems), respectively. The resulting high-quality libraries were sequenced in parallel on Illumina HiSeq™ 2000 system (Beijing Ori-Gene Science and Technology Co., Ltd, Beijing, China).

## 2.3. Quality control of raw data and mapping reads to the genome

The raw data were trimmed and filtered by removing adapter reads, poly-N reads, reads with length shorter than 60 bp and low-quality reads containing uncertain bases 'N' to obtain the clean reads. Clean reads were mapped to the *O. aries* reference genome (Oar_v4.0 genomic dataset, ftp://ftp.ncbi.nlm.nih.gov/genomes/all/GCF/000/298/735/GCF_000298735.2_Oar_v4.0/GCF_000298735.2_Oar_v4.0_genomic.fna.gz) using Bowtie2 (2.2.4) software from Tophat2 (2.0.10) [23].

## 2.4. Identification and screening of DEGs between HMTs and CMMTs

The mRNA abundance was measured by reads per kilobase per million reads (RPKM) using the Cufflinks packages as described previously [24,25]. Differential expression analysis of genes between HMTs and CMMTs was implemented using the Cuffdiff [26]. Genes with indexes $|\log_2(\text{fold change})| > 1$ and FDR $< 0.05$ were seen as DEGs.

## 2.5. Functional annotation and pathway analysis of DEGs

GO annotation and KEGG pathways analyses of DEGs were implemented by DAVID 6.8 (https://david.ncifcrf.gov/) [27] and KOBAS 3.0 online program (http://kobas.cbi.pku.edu.cn/index.php) [28], respectively. GO terms and KEGG terms with a criterion-corrected $p < 0.05$ were seen as significantly enriched by DEGs. GO terms with $p < 0.01$ in Immune System Process were also enriched by Cytoscape application ClueGO based on *O. aries* GO database.

## 2.6. Construction of the co-expression and protein–protein interaction network for immunity-related DEGs

Pairs with a correlation coefficient greater than 0.99 or less than $-0.99$ and $p < 0.05$ were considered as co-expressed correlations to establish the co-expression network, and then the network was visualized using Cytoscape 3.7 software [29]. Pearson's correlation coefficient for each pair of genes was calculated using the cor () function, and $p$-value was calculated using the cor.test function in R language. Genes were represented using nodes and the interactions between two genes were represented by edges. The protein–protein interaction (PPI) network for immune response-related DEGs was constructed using the Search Tool for the Retrieval of Interactive Genes (STRING) database (https://string-db.org/) to obtain interacting genes. Proteins indicated nodes and the interactions between two proteins indicated edges.

## 2.7. Analyses of differential splicing genes

To screen differential splicing genes (DSGs) in CMMTs and HMTs, the differential alternative splicing (AS) events from RNA-seq data were examined by rMATS [30]. A FDR $< 0.05$ was considered as the threshold to judge the significance of DSGs.

## 2.8. Validation of RNA-seq results by quantitative real-time PCR

To verify reliability of the transcriptomic profiling data, quantitative real-time PCR (qRT-PCR) was performed for 10 randomly selected DEGs (seven upregulated genes: *CD19, CD79A, CD79B, CXCR2,*

*IL1R2*, *S100A8* and *TLR2*; three downregulated genes: *EPHX2*, *KLF4* and *RTN4RL1*), depending on the operating manual. *GAPDH* was used as a reference gene in quantitative analysis. The purity and concentration of extracted RNA were assessed using an ultra-micro ultraviolet spectrophotometer (Implen, Germany), and the integrity of RNA specimens were detected with 1% agarose gels. The first-strand cDNA was synthesized from an equal amount of RNA (250 ng) with oligo(dT)$_{18}$ primers using TransScript All-in-One First-Strand cDNA Synthesis SuperMix (TransGen Biotech, China) according to kit instructions. The randomly selected 10 DEGs were amplified (Stratagene Mx3005P, Agilent Technologies, USA) by an optimized two-step, namely one cycle of 94°C for 30 s, 40 cycles of 94°C for 5 s and annealing temperature (60.5°C) for 30 s, or three-step, namely one cycle of 94°C for 30 s, 40 cycles of 94°C for 5 s and annealing temperature (less than 60°C) for 15 s and 72°C for 10 s. The amplification reaction system (20 µl) comprised 1 µl of cDNA, 0.4 µl of forward primer, 0.4 µl of reverse primer, 10 µl of 2× TransStart® Tip Green qPCR SuperMix (TransGen Biotech, China) and 8.2 µl of ddH$_2$O. The relative mRNA expression levels of target genes were normalized to the expression of *GAPDH*, and then were calculated using the $2^{-\Delta\Delta C_t}$ method [31]. The qRT-PCR primers used in this study are listed in the electronic supplementary material, table S1.

## 2.9. Western blot analysis

Total protein was extracted from each sample using a RIPA protein extraction kit (Solarbio Bio-Technology Co., Ltd, Beijing, China) according to the kit manual. Denatured protein samples of equal volume were separated by 12% sodium dodecyl sulfate–polyacrylamide gel electrophoresis and then transferred to polyvinylidene difluoride blotting membranes (Beyotime Biotechnology, Shanghai, China). Transmembrane transfer was incubated with a rabbit anti-RTN4RL1 polyclonal antibody (1 : 500, Bioss Biotechnology Co., Ltd, Beijing, China), anti-TLR2 polyclonal antibody (1 : 500, Bioss Biotechnology Co., Ltd) and anti-β-actin polyclonal antibody (loading control) (1 : 1500, Bioss Biotechnology Co., Ltd) at 4°C overnight, respectively. After washing in PBST, membranes were incubated with goat anti-rabbit IgG/HRP antibody (1 : 5000, Bioss Biotechnology Co., Ltd). For more detailed information about the operational methods, see our previous report [32].

## 2.10. Haematoxylin and eosin staining

Sections from HMTs and CMMTs were stained with H&E, dehydrated, dewaxed with conventional histological methods as described previously [32,33] with some modifications. Sections were observed under a microscope (Sunny Optical Technology Co. Ltd, Ningbo, China), and photographed using ImageView (Sunny Optical Technology Co. Ltd).

## 2.11. Immunocytochemistry

Immunocytochemistry was performed as previously described [30]. The sections were incubated with rabbit anti-CD19 (1 : 100; Bioss, China), anti-CD79B (1 : 100; Bioss), anti-RTN4RL1 (1 : 100; Bioss) and anti-TLR2 (1 : 100; Bioss) polyclonal antibody, respectively. PBS replaced the primary antibody as the negative control. Sections were observed and captured using Sunny EX31 biological microscope (Sunny Optical Technology Co. Ltd) and ImageView software (Sunny Optical Technology Co. Ltd), respectively.

## 2.12. Data statistics and analysis

All statistical analyses were conducted with an independent-samples *t*-test using SPSS 19.0 software (SPSS Inc., Chicago, IL, USA). The results were presented as mean ± s.d. Values of $p < 0.05$ were considered as statistically significant, and values of $p < 0.01$ were considered as statistically extremely significant.

# 3. Results

## 3.1. Comparison of morphological differences between HMTs and CMMTs

HE staining showed that the sections from HMTs exhibited a few connective tissues including interlobular and intralobular connective tissues and acinar lumina with little cellular debris

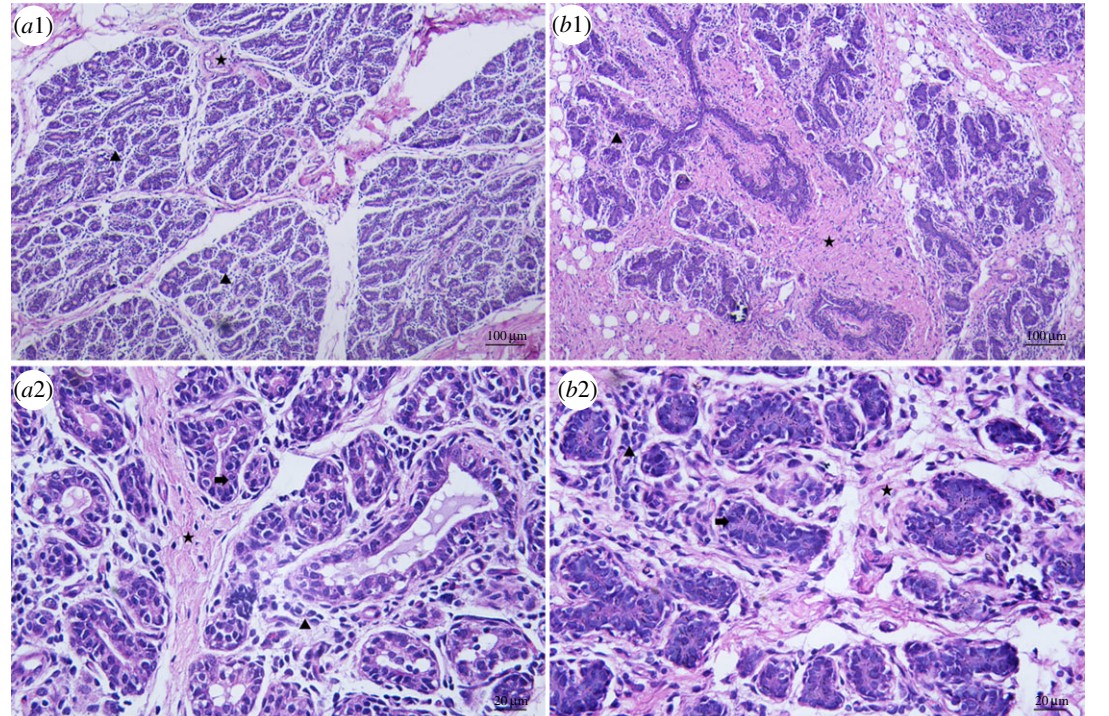

**Figure 1.** Morphologic characteristics between HMTs and CMMTs by H&E. (*a*1) and (*a*2) were the morphology of HMTs under microscope at 100× and 400× magnification, respectively, and (*b*1) and (*b*2) were the morphology of CMMTs under microscope at 100× and 400× magnification, respectively. A black arrow indicates mammary epithelium, a star indicates interlobular connective tissues and a triangle indicates intralobular connective tissues.

(figure 1*a*1,*a*2), whereas the sections from CMMTs were characterized by connective tissue hyperplasia, especially interlobular connective tissues, reduced mammary gland lobules and acinar lumina, mammary epithelium exfoliation, infiltration of abundant inflammatory cells such as neutrophils and lymphocytes, and the presence of a large amount of cellular debris in the acinar chamber (figure 1*b*1,*b*2).

## 3.2. Data analysis from RNA-seq

RNA-seq data for the HMTs and CMMTs groups are shown in table S2. The number of raw reads from CMMTs and HMTs, on average, for both was above 60 million (M). After filtering, all of the clean reads were greater than 92% of the raw data. Of the total clean reads, on average, 70.55% clean reads from the CMMTs group and 77.73% clean reads from the HMTs group were mapped to the ovine reference genome, respectively. The unique match ratios were 66.54% and 73.87%, and multiple match ratios were 4.01% and 3.86%, for the CMMTs and HMTs groups, respectively.

## 3.3. Gene expression analysis

The transcriptome data revealed that 10 924 genes were co-expressed in HMTs versus CMMTs, 420 of which were significantly differentially expressed in CMMTs based on the screening standard of DEGs (absolute $\log_2$FC > 1, FDR < 0.05). Of the 420 DEGs, 316 genes were markedly upregulated, while 104 genes were markedly downregulated in CMMTs (figure 2*a*,*b*). More information on these DEGs is provided in table S3. The cluster analyses of 43 immunity-related DEGs between HMTs and CMMTs are presented in figure 2*c*. The expression levels of the 37 DEGs were significantly upregulated, whereas the other four DEGs were significantly downregulated in CMMTs, when compared with HMTs.

## 3.4. GO annotation and KEGG pathway enrichment analysis of DEGs

GO annotation revealed that most of the 420 DEGs were mainly involved in biological processes related to immune and inflammatory response, such as B-cell receptor (BCR) signalling, innate immune

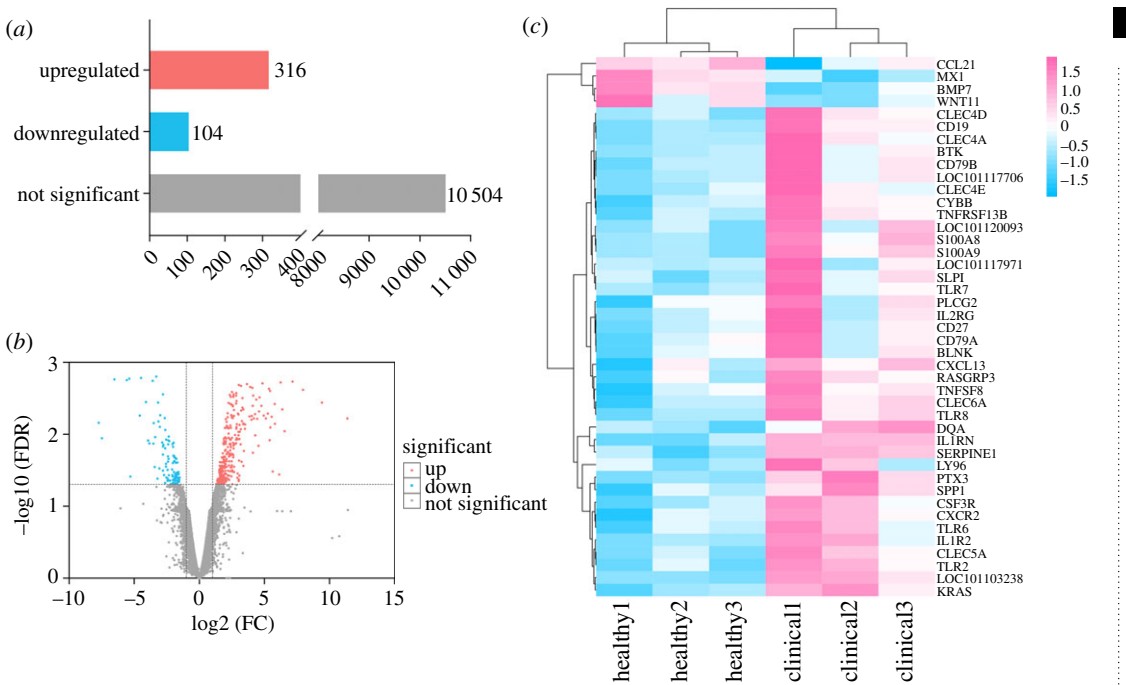

**Figure 2.** Gene expression results between HMTs and CMMTs by RNA-seq. (*a*) The histogram shows the number of DEGs and not significant change genes in CMMTs compared with HMTs. (*b*) The volcano plot of co-expressed genes between HMTs and CMMTs. It comprises FDR($-\log_{10}$(FDR), *y*-axis) and fold change of expression levels of genes between CMMTs and HMTs ($\log_2$FC, *x*-axis). Red indicates upregulated, blue indicates downregulated and grey indicates no significantly changed expression levels in CMMTs compared with HMTs. (*c*) The clustering heatmap of immune response-related DEGs between HMTs and CMMTs. Each column represents a sample, and each row represents the expression levels of a single gene in various samples. Sample names indicate the mammary glands of healthy (healthy1, healthy2 and healthy3) and clinical mastitis (clinical1, clinical2 and clinical3).

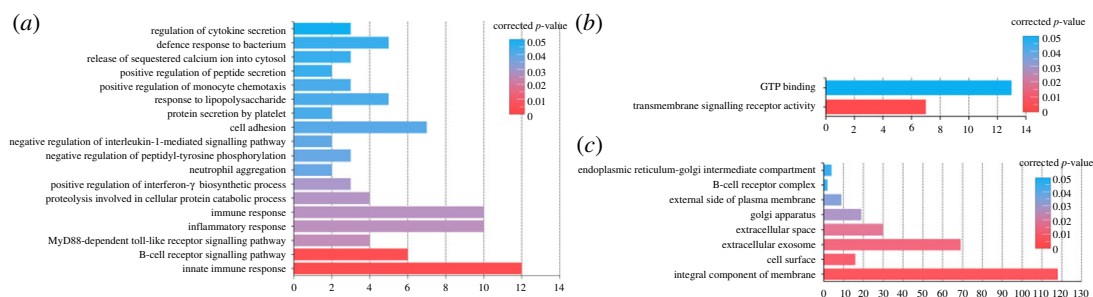

**Figure 3.** GO annotation results of DEGs. (*a*) Biological processes, (*b*) molecular function and (*c*) cellular components.

response, response to lipopolysaccharide, defence response to bacterium and MyD88-dependent toll-like receptor (TLR) signalling pathway (figure 3*a*). On the molecular function, the DEGs were significantly enriched in two terms, including GTP binding and transmembrane signalling receptor activity (figure 3*b*). On the cellular components, DEGs were associated with eight terms, such as integral component of membrane, extracellular exosome, BCR complex and extracellular space (figure 3*c*). The detailed information of GO annotation results of DEGs from three classes including the biological processes, molecular function and cellular components is summarized in table S4.

KEGG enrichment results are shown in figure 4. The 15 upregulated genes were significantly enriched in pathway related to phagosome, such as *TLR2*, *TLR6*, *C3*, *CYBB*, etc. The 13 DEGs were engaged in cytokine–cytokine receptor interaction, 11 of which were upregulated, such as *IL1R2*, *CXCR2*, *CXCL13* and *CD27*, whereas two were downregulated, including *CCL21* and *BMP7*. The nine upregulated genes were engaged in BCR signalling, such as *CD19*, *CD79A*, *CD79B*, *BTK* and *BLNK*, etc. The six upregulated genes were significantly enriched in TLR signalling pathway, such as *TLR2*, *TLR6*, *TLR7*,

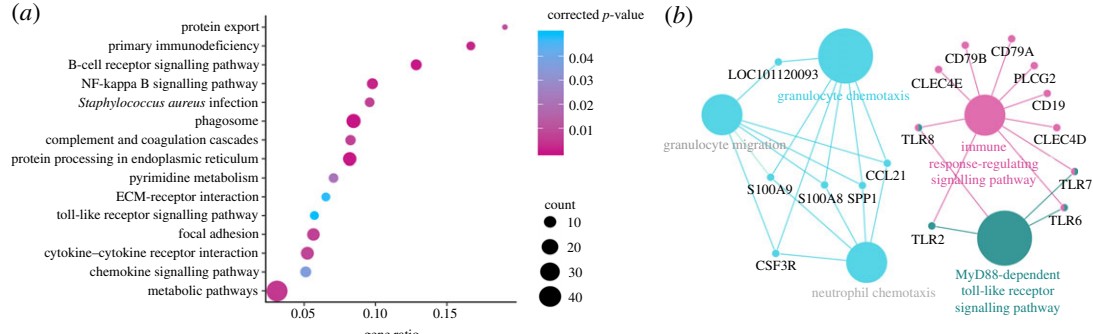

**Figure 4.** Pathway enrichment analyses of DEGs. (*a*) Scatter plot of KEGG pathways from DEGs and (*b*) immune systems mediated by immune-related DEGs.

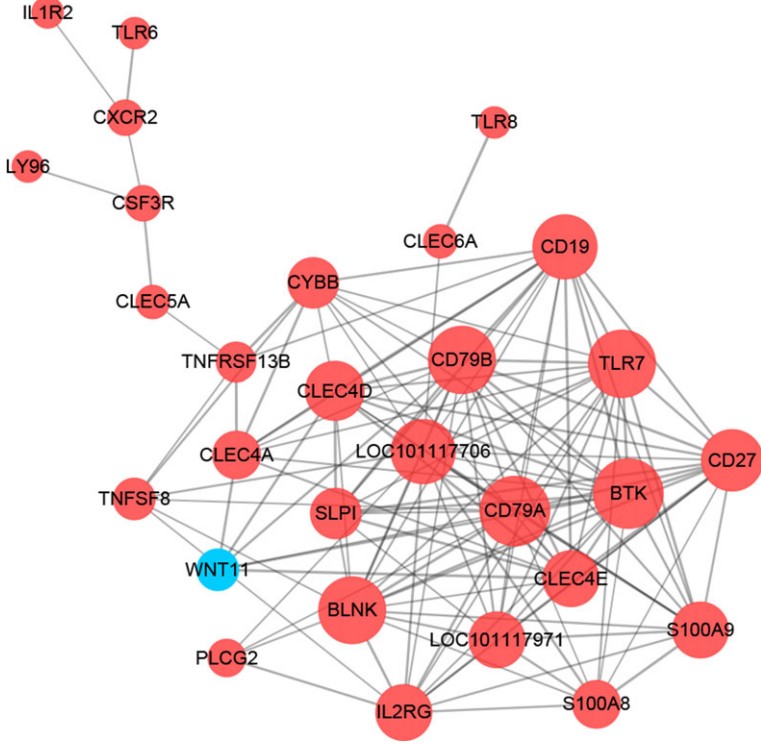

**Figure 5.** Gene co-expression network analyses of DEGs. A red node indicates upregulated gene, while the blue node indicates a downregulated gene in CMMTs rather than HMTs.

*TLR8*, etc. The six upregulated genes were associated with primary immunodeficiency, including *CD19*, *TNFRSF13B*, *IL2RG*, *CD79A*, *BLNK* and *BTK*. More descriptions of KEGG enrichment results of DEGs are summarized in table S5.

## 3.5. Co-expression network and PPI network

The differential co-expression network of DEGs associated with immune response was constructed. As shown in figure 5, there are 29 nodes (28 upregulated and 1 downregulated genes) and 133 edges in the network. From the analysis, the hub genes with a higher degree were *BTK* (Bruton tyrosine kinase, degree = 17), *CD79A* (degree = 17), *BLNK* (B-cell linker, degree = 16), *TLR7* (toll-like receptor 7, degree = 16), *CD79B* (degree = 17) and *CD19* (degree = 15). The PPI network of immune response-related DEGs in CMMTs are presented in figure 6. There were 109 edges and 34 nodes (31 upregulated and 3 downregulated genes) in the network. Based on the PPI network, *CD19* (degree = 11), *BTK* (degree = 10), *TLR2* (degree = 10), *TLR7* (degree = 10) and *TLR8* (degree = 7) were the top five hub genes.

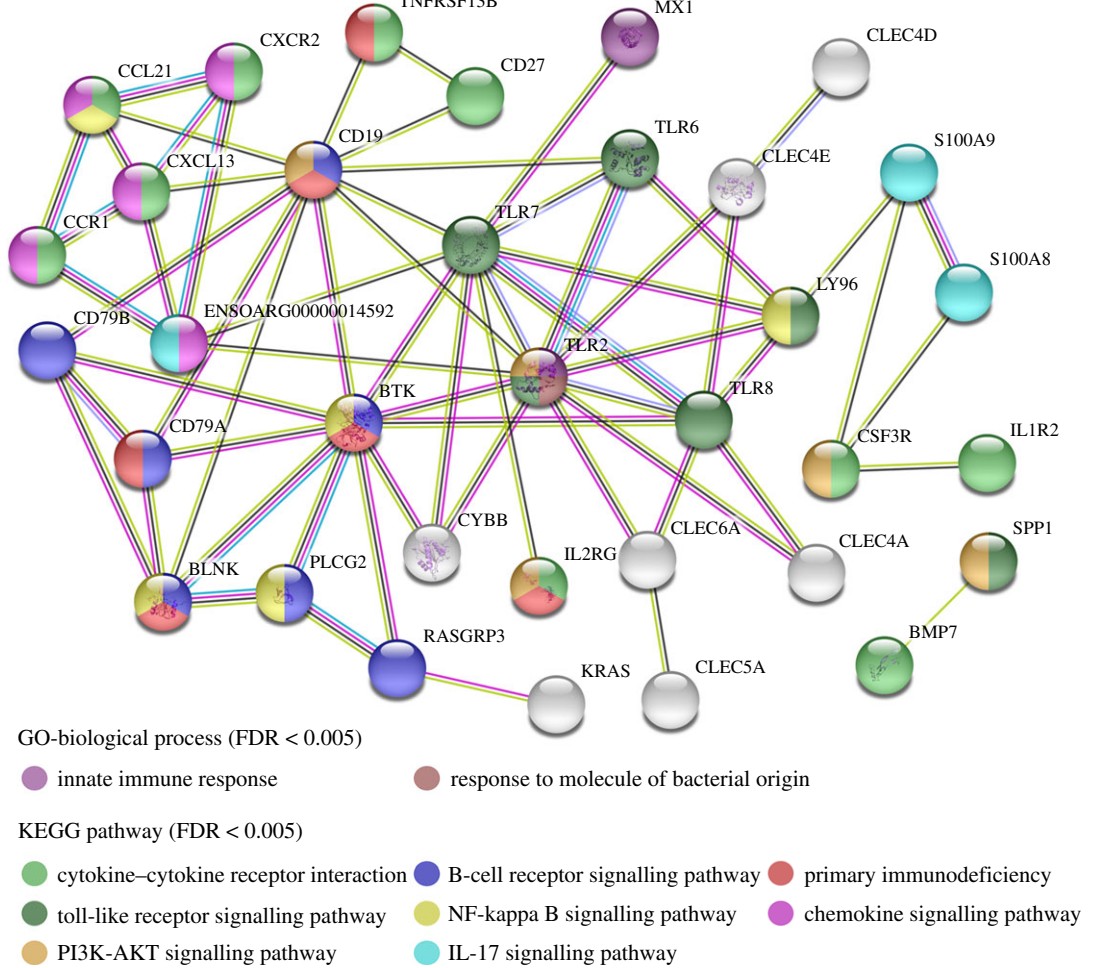

**Figure 6.** PPI network analyses of identified immune response-related DEGs.

## 3.6. Identification of genes with differential splicing events

In the current study, we found the five main AS types including skipped exon (SE), alternative 5′ splicing site (A5SS), alternative 3′ splicing site (A3SS), mutually exclusive exons (MXE) and retained intron (RI). Based on a threshold of FDR < 0.05, there were 59 DSGs with SE such as *ERBB2*, *RAB1A*, *XDH* and *ATG4C*, 36 DSGs with RI such as *RNF121*, *PINK1*, *DHX30* and *KDM5C*, 24 DSGs with A3SS such as *RPL22L1*, *TNFRSF13B* and *BMP1*, 19 DSGs with MXE such as *OGDH*, *TNC* and *VPS51*, and 13 DSGs with A5SS such as *MYL6*, *BRD7* and *ALKBH6* in comparisons of CMMTs and HMTs (figure 7).

## 3.7. Validation of RNA-seq data by qRT-PCR

As shown in figure 8*a*, mRNA expression levels of *CD19*, *CD79B*, *S100A8*, *TLR2*, *CD79A*, *CXCR2* and *IL1R2* were significantly higher, while *EPHX2*, *KLF4*, and *RTN4RL1* mRNA expression levels were significantly lower in CMMTs than those in HMTs ($p < 0.01$). qRT-PCR results showed that expressions of the selected 10 DEGs were consistent with the changing trends from RNA-seq data, but there were differences in fold changes between RNA-seq and qRT-PCR (figure 8*b*).

## 3.8. Expression and localization of proteins encoded by individual DEGs

The relative expression levels of proteins encoded by randomly selected two DEGs (*TLR2* and *RTN4RL1*) were examined by western blot. The results indicated that the expression level of RTN4RL1 protein was significantly decreased ($p < 0.05$), whereas TLR2 protein was significantly increased in CMMTs, when compared with HMTs ($p < 0.05$) (figure 9*a*).

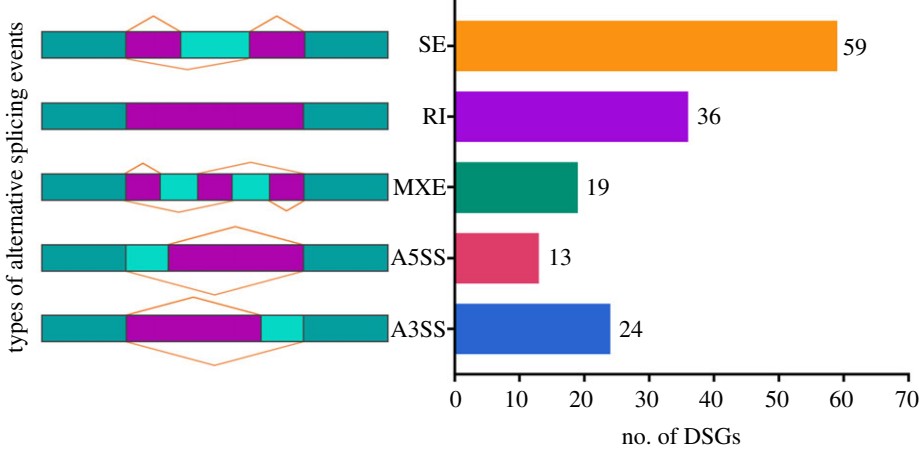

**Figure 7.** Identification of DSGs regulated by SE, RI, MXE, A5SS and A3SS. FDR < 0.05.

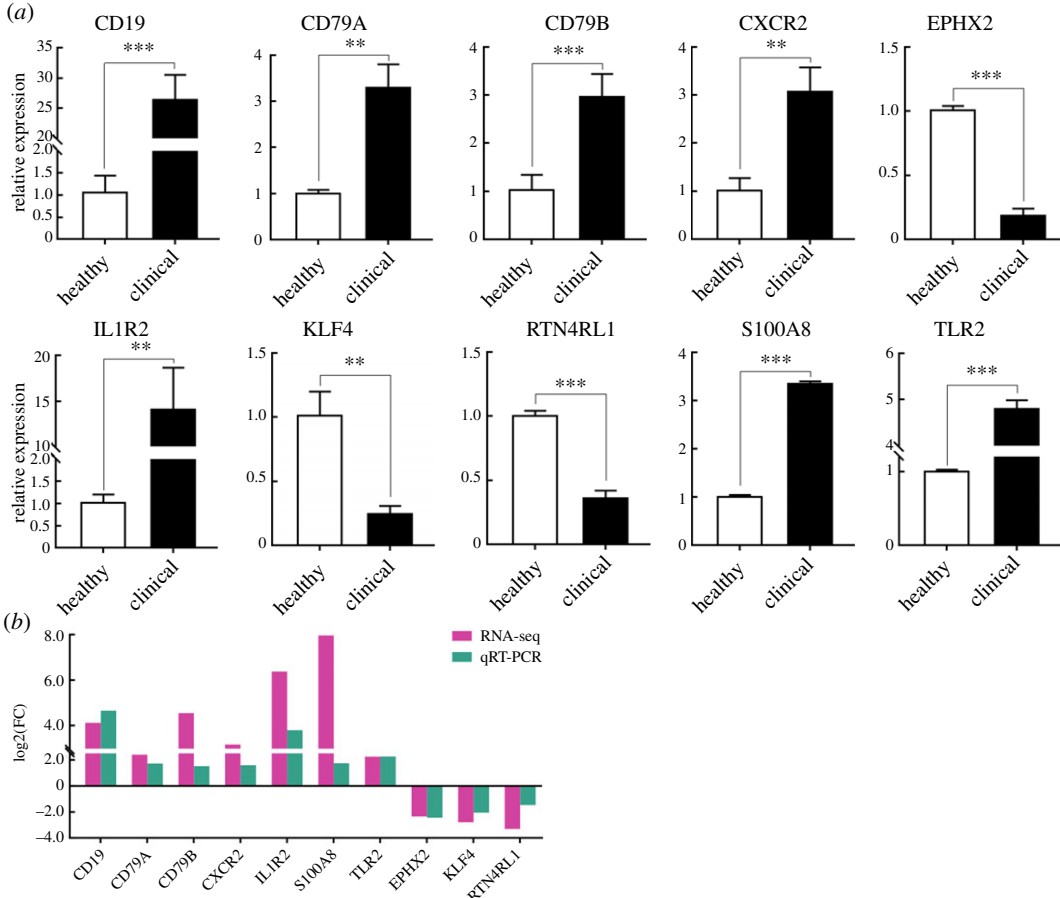

**Figure 8.** Expression patterns of 12 randomly selected DEGs. (*a*) mRNA expression levels were confirmed by qRT-PCR. (*b*) Comparison of fold change of mRNA expression in CMMTs versus HMTs from RNA-seq and qRT-PCR. *GAPDH* was used as control. Three biological replicates with three technical replicates each were used. Data were presented as mean ± s.d. in the graphs. \*\**p* < 0.01, \*\*\**p* < 0.001.

Immunohistochemical staining showed that the distribution and intensity of positive signals for the same protein in HMTs and CMMTs varied. CD19 protein was located in mammary epithelium and intralobular connective tissues in CMMTs and HMTs with stronger positive signals in CMMTs when compared with HMTs (figure 9*b*). CD79B protein was strongly located in mammary epithelium of CMMTs, but it was restrained in mammary epithelium of HMTs (figure 9*b*). TLR2 protein was located in mammary epithelium from both CMMTs and HMTs, with strong positive expressions examined in

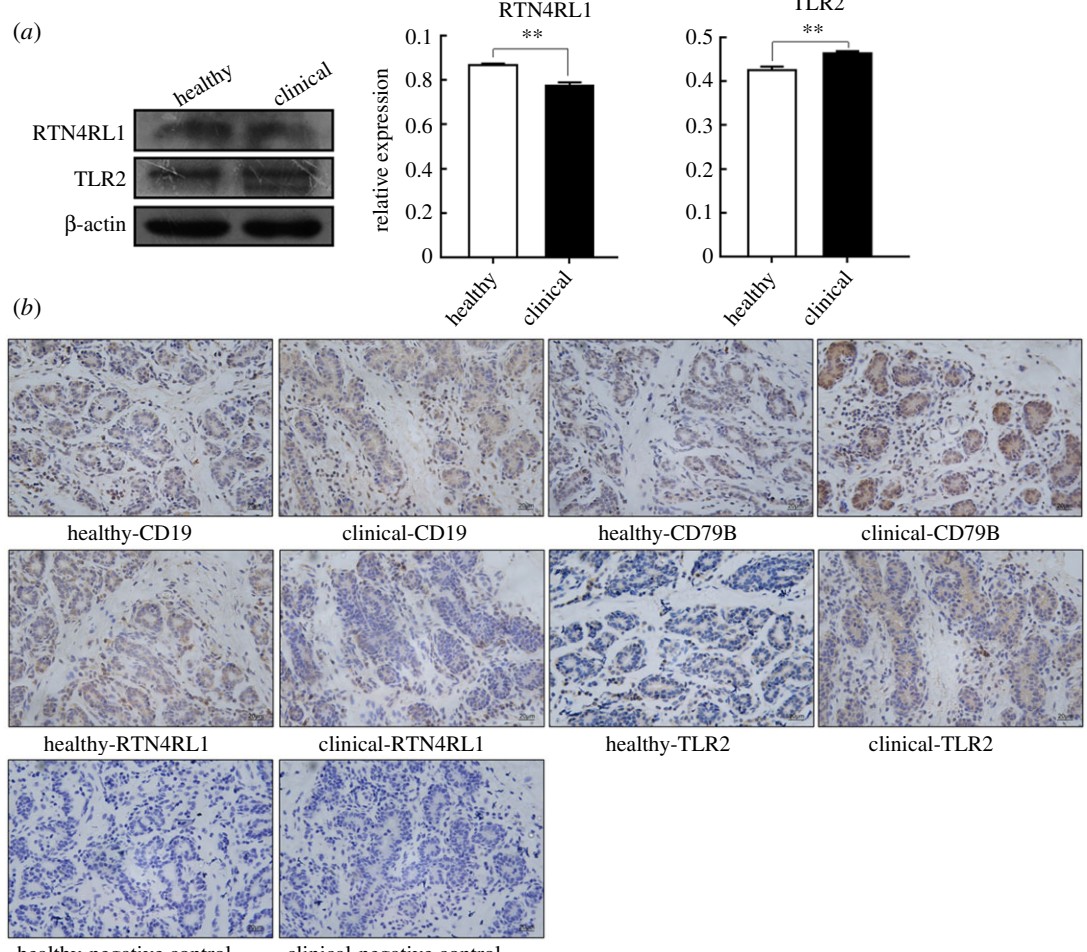

**Figure 9.** Expression patterns of proteins encoded by some of DEGs. (*a*) Coding protein expressions of two randomly selected DEGs were confirmed by western blot. (*b*) Immunohistochemical staining of proteins encoded by five DEGs. PBS was used as the negative control. All the representative images from sections were at the same magnification (400×). **$p < 0.01$.

CMMTs (figure 9*b*). RTN4RL1 protein was located in mammary epithelium and intralobular connective tissues from both CMMTs and HMTs, with a strong positive reaction in HMTs but not CMMTs (figure 9*b*).

## 4. Discussion

During mastitis, the mammary microenvironment, including epithelial cells and stromal components such as fibroblasts, endotheliocytes and immunocytes, is greatly changed [34]. Morphologically, our observations revealed that mammary glands from ewes with clinical mastitis were characterized by massive interlobular connective tissue proliferation in the stromal regions and inflammatory cell infiltration. The results are similar to the observation of Chopradewasthaly *et al*. [21], who reported that mastitic sheep mammary gland infected with *Mycoplasma agalactiae* appears to exhibit significant inter- and intra-lobular connective tissue hyperblastosis. Our results suggest that the histological structures of mammary glands of ewes with clinical mastitis are obviously changed by the invasion of pathogens compared with healthy ones.

In the present study, clean reads compared with reference genome results showed that above 68 million reads in CMMTs were accomplished, but nearly 30% of the clean reads could not be mapped to the reference genome. This is probably due to the lack of genome information [35]. Here, we screened 420 genes with no less than twofold change in mRNA abundance based on FDR < 0.05 in clinically mastitic mammary glands in comparison with healthy ones, and mRNA expression levels of many immune response-related genes were significantly upregulated in mammary glands with clinical mastitis. Based on the comparisons with previous transcriptomic studies on mastitis in sheep or cattle, there were many similar (table S6) or homologous genes relating to immune and

inflammation response identified in this work. After bacterial infection, the immune system in the mammary gland is activated, and massive amounts of leucocytes are migrated from the blood to the site of infection of the mammary gland to participate in inflammatory processes and responses to invading pathogens [36,37]. These indicate upregulated genes related to immunity play crucial roles in host defence. In previous transcriptome analyses from dairy sheep, several genes relating to mastitis resistance or innate immunity including PLXNC1, CCNYL1 and PPP4R2 were identified by Banos et al. [38], which were in agreement with our transcriptome data. These might help to better understand the disease-resistant mechanisms in sheep. In a previous RNA-seq report, nine immune-related genes such as CXCL13, LYZ, CD79B, CD19/86, S100A8/12, DQA1 and CCL19 were upregulated in mastitic sheep mammary tissues experimentally infected with M. agalactiae [21]. Similarly, in this study, some of these genes (CXCL13, LYZ, CD79B, CD19 and S100A8) as well as some genes within the same family (CD27/53/55, S100A9, DQA and CCL21) were upregulated in CMMTs when compared with HMTs. Also, consistent with our findings, four upregulated genes associated with immunity—S100A8, S100A9, SERPINE1 and IL1RN—in goat mammary epithelial cells infected with M. agalactiae and cow mammary tissues infected with coagulase-positive Staphylococci were also reported by two other papers concerning transcriptomic studies [39,40]. Furthermore, our qRT-PCR data also showed that mRNA levels for CD19, CD79A, CD79B and S100A8 were significantly higher in CMMTs than those in HMTs, and immunolocalization results indicated that intense positive signals for CD79B and CD19 were detected in mammary epithelium of CMMTs. CD19, CD79A and CD79B belong to the molecular mechanisms that are involved in BCR signalling. Previous research has proven that BCR signalling is essential for the development and functional maintenance of B cells, and this has been confirmed as a central driver for the development and evolution of various B-cell malignancies [41,42]. Unfortunately, there are still relatively rare studies on BCR signalling in ovine mastitis.

Moreover, as per previous transcriptome reports on cow mammary tissues infected with coagulase-positive Staphylococci by Kosciuczuk et al. [40], we also found that another 28 genes (such as CXCR2, BLNK, CYBB and CLEC6A) with higher expressions in CMMTs were implicated in immune and inflammation response. According to further functional annotation and pathway analyses, these genes are mainly involved in biological processes and/or pathways such as innate immune response, primary immunodeficiency, inflammatory response and neutrophil aggregation. For example, CXCR2, a key chemokine receptor that is a member of the G protein-coupled receptors superfamily, is essential for inflammatory cells migration to infection sites [43,44]. In mastitic cows, CXCR2 mRNA expression is obviously higher in milk neutrophils compared with healthy ones, which has also been described in prior publications by Alhussien & Dang [45] and Alhussien et al. [46]. Additionally, previous literature documents that the CXCR2 gene polymorphism is closely interrelated with bovine mastitis susceptibility or resistance [43,47]. However, whether CXCR2 gene polymorphism is related with clinical mastitis in sheep still needs to be further investigated and explored.

In agreement with the previous reports on mastitic transcriptome analyses from goat mammary epithelial cells [39] and cow mammary tissues [48], we also examined two immune-associated genes (TLR2 and PTX3) that were significantly upregulated in CMMTs when compared with HMTs. They mainly engaged in GO biological processes including innate immune response, MyD88-dependent TLR signalling pathway and regulation of cytokine secretion, along with pertaining to KEGG pathways, including phagosome and TLR signalling pathway. TLR2 is an important member of the TLRs family that plays a crucial regulatory role in immune systems during diseases [49,50], and mastitis is no exception [51,52]. Just like our RNA-seq results, significantly increased TLR2 expressions at mRNA and protein levels were detected by qRT-PCR and western blot, and TLR2 proteins with stronger positive signals were observed in CMMTs when compared with HMTs. Strong TLR2 mRNA and protein expressions were also reported in mammary glands infected with S. aureus from small-tail Han sheep [53] and mice [54]. These reveal that TLR2 plays an essential role in the immune response of mammary epithelium and can be used as a candidate gene for clinical sheep mastitis, which deserves further study.

In the present study, we also screened 19 genes with differential expressions (18 upregulated genes, such as PRKCB and BTK, and one downregulated gene, AGTR1), which were most associated with immune response and defence, such as defence response to bacterium, inflammatory response and positive regulation of monocyte chemotaxis, and these results were in accordance with the previous transcriptomic study on cow mammary tissues infected with coagulase-negative Staphylococci reported by Kosciuczuk et al. [40]. In addition, in coincidence with the results of transcriptomic analyses on mammary tissues derived from cows naturally infected with S. aureus mastitis [55], we found in the

present study that one immune-related gene *GPR68* and seven other genes such as *ADAMTS12*, *COL1A1* and *LRRC8C* were upregulated, while two immune-related genes (*MX1* and *AOX1*) and 11 other genes such as *GATSL3*, *UBXN11* and *FHOD1* were downregulated in the infected group but not in the healthy group. Moreover, in addition to *S100A9*, we also identified two other genes (*SLAMF7* and *C3*) with higher expressions in CMMTs, which was in accordance with reports on transcriptome of bovine mammary tissues infected with *Streptococcus uberis* from Swanson *et al*. [56]. *C3* is recognized as a key molecule of the complement system that regulates host defence against pathogens during mastitis [57]. All these findings demonstrate that host immune defence mechanism against bacterial infection may be shared partially during mastitis in sheep, goat and cow.

AS is an important regulatory phenomenon that widely exists in eukaryotes for regulating various biological processes [58]. For instance, it participates in the initiation and progression of some diseases through generating multiple transcripts from the same primary RNA sequence [58]. During mastitis, many studies have showed that expressions of the same gene with different splicing patterns are variable, to serve their critical roles in mastitis resistance or susceptibility [55,59–61]. In a transcriptome study on mammary glands from cows with *S. aureus* mastitis, most genes with AS events were implicated in immune response as reported by Wang *et al*. [55]. Our RNA-seq data revealed that the numbers of DSGs with SE, A5SS, A3SS, MXE and RI were 59, 13, 24, 19 and 36, respectively, in potential AS event comparisons between CMMTs and HMTs (FDR < 0.05). Of these, many DSGs were associated with immune response, such as *TNFRSF13B* and *CTNNB1* with A3SS, *IRF7* and *LAMB3* with RI, *CAMK2D*, *CSNK1A1* and *CAMK2D* with A5SS, *GSN*, *ATG4C* and *ERBB2* with SE, as well as DSGs associated with metabolism, such as *PHYKPL*, *ALG3* and *NT5C2* with RI, *SAT1*, *NDUFS2*, *BCKDHB*, *SAT1*, *GSS*, *AOX1* and *XDH* with SE, *DGKQ* and *INPP5 K* with A5SS, and *GALT*, *KDSR*, *OGDH* and *PLPP1* with MXE. Some of the DSGs associated with various types of AS events were also identified, which included DSGs involved in immune response, such as the *ADGRF5* gene with SE and A3SS, the *TNC* gene with SE and MXE, and *RNF121* with RI and MXE, along with DSGs involved in cell proliferation, such as the *HNRNPDL* gene with RI, SE and A3SS, and *RBM3* with SE and A3SS. For instance, *SBNO2*, an inflammatory response-related gene, a previous report indicates that it exhibited RI patterns in cow mammary tissues with healthy and S. *aureus* mastitis [55]. Herein, we also identified *SBNO2* was a differential splicing gene, but it exhibited A3SS patterns in HMTs and CMMTs, suggesting it may have a different regulatory role in the inflammatory response of mammary gland from mastitic sheep compared with cows. Generally, these DSGs provide a wealth of material for further mechanism studies on regulation of gene expression and host defence during mastitis in sheep.

# 5. Conclusion

This is the first report investigating the transcriptome changes of mammary glands from meat sheep with clinical mastitis caused by natural infection. A total of 420 DEGs were identified between CMMTs and HMTs. We mainly screened some of the immunity-related DEGs in ovine mammary glands during clinical mastitis, and of these, most expressions are upregulated in CMMTs compared with HMTs. Many genes with differential splicing events were also examined, out of which some were implicated in immunity. Moreover, more intensive positive signals from three immunity-related proteins were observed in mammary epithelium of CMMTs. We conclude by speculating that these three DEGs may contribute to regulate the immune response of mammary epithelium during clinical mastitis. We will further dissect the functions of some of the DEGs screened in this study, so as to provide a valuable reference for immune-related genes as biomarkers in the early diagnosis and preventive treatment of clinical ovine mastitis.

Ethics. All animals were managed according to the animal care and experimental procedure guidelines established by the Ministry of Science and Technology of the People's Republic of China (Approval No. 2006-398), and approved by the Animal Care Committee of Gansu Agricultural University.

Data accessibility. Our data are deposited at GenBank (accession NO. SRP174379, https://www.ncbi.nlm.nih.gov/search/all/?term=SRP174379). Data are available from the Dryad Digital Repository at: https://doi.org/10.5061/dryad.7mg3d80) [62].

Authors' contributions. Y.M. and T.L. conceived and designed the work. T.L. and J.G. assisted in collecting and compiling the resource materials and in manuscript preparation. T.L. conducted the research work, and then wrote the manuscript. Y.M. and X.Z. revised the manuscript. All authors read and approved the final manuscript.

Competing interests. We declare that we have no competing financial interests.

Funding. The research was funded by the discipline construction fund project of Gansu Agricultural University (grant no. GSAU-XKJS-2018-021).

Acknowledgements. We thank Pingchang Hu Sheep Breeding Base (Lintao, Gansu, China) for providing the experimental animals and Ori-Gene Science and Technology Co., Ltd (Beijing, China) for help with the RNA-sequencing.

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
