## [Reviewer comments · Royal Society Open Science]

Review History

RSOS-181604.R0 (Original submission)

Review form: Reviewer 1

Is the manuscript scientifically sound in its present form?

Yes

Are the interpretations and conclusions justified by the results?

Yes

Is the language acceptable?

Yes

Is it clear how to access all supporting data?

Yes

Do you have any ethical concerns with this paper?

No

Have you any concerns about statistical analyses in this paper?

No

Recommendation?

Accept as is

Comments to the Author(s)

The manuscript describes transcriptomic changes in mammary glands of sheep affected with clinical mastitis compared to healthy mammary tissues. The study resulted in the typical long list of DEGs, that were subsequently grouped into a meaningful functional categorization. Validation of RNA-seq results was done for a few genes with RT-qPCR and immunohistochemistry. The study was well-conducted and as transcriptome data of clinical mastitis in sheep are yet not available it represents an important addition to literature.

Review form: Reviewer 2

Is the manuscript scientifically sound in its present form?

No

Are the interpretations and conclusions justified by the results?

No

Is the language acceptable?

Yes

Is it clear how to access all supporting data?

Yes

Do you have any ethical concerns with this paper?

No

Have you any concerns about statistical analyses in this paper?

Yes

Recommendation?

Major revision is needed (please make suggestions in comments)

Comments to the Author(s)

This is an interesting manuscript studying the transcriptome signature of clinical mastitis in sheep.

Although the manuscript is well written, the sample size is small and the authors used very mild significant thresholds. Stricter significant thresholds-at least adjusting for multiple testing- should be performed for the Differential Expression analysis. All the analysis related to co-expression, pathway and network analysis etc should be repeated using only the DE results after adjusting for multiple testing using for example FDR correction.

Moreover, the whole discussion should be re-written in a more focused way with in depth comparisons among the present studies and previous studies on the topic. The authors also miss recent publications on the topic for example

<https://bmcgenomics.biomedcentral.com/articles/10.1186/s12864-017-3982-1>

Material and Methods:

The authors talk about natural infection-clinical mastitis without clarifying which was the causing agent. Did they perform microbiological analysis to identify the responsible pathogen? Was the responsible pathogen the same for all three cases?

Line 22: The current reference genome version is *Ovis aries* v3.1. The authors mention that they are using Oar_v4.0 genomic dataset. Could the authors give some more details on these?

Line 36: I assume that the $P < 0.05$ is the uncorrected P value and not the adjusted after FDR correction. I am wondering how many genes remain significantly differentially expressed after adjusting for multiple testing using FDR correction.

A $P < 0.05$ is a very mild threshold and the results could be false positive. This should be amended and the genes which are presented as differentially expressed should be the ones which remain significant after correcting for multiple testing.

Did you find in the two groups any differentially spliced genes?

Decision letter (RSOS-181604.R0)

26-Mar-2019

Dear Dr Li,

The editors assigned to your paper ("Digital gene expression analyses of mammary glands from meat ewes naturally infected with clinical mastitis") have now received comments from reviewers.

While one reviewer is positive about publication of your paper, the other reviewer raises some substantive comments on significance thresholds and the analysis of expression data. It will be important to address carefully these comments in your revision.

We would like you to revise your paper in accordance with the referee's suggestions which can be found below (not including confidential reports to the Editor). Please note this decision does not guarantee eventual acceptance.

Please submit a copy of your revised paper before 18-Apr-2019. Please note that the revision deadline will expire at 00.00am on this date. If we do not hear from you within this time then it will be assumed that the paper has been withdrawn. In exceptional circumstances, extensions may be possible if agreed with the Editorial Office in advance. We do not allow multiple rounds of revision so we urge you to make every effort to fully address all of the comments at this stage. If deemed necessary by the Editors, your manuscript will be sent back to one or more of the original reviewers for assessment. If the original reviewers are not available, we may invite new reviewers.

When submitting your revised manuscript, you must respond to the comments made by the referees and upload a file "Response to Referees" in "Section 6 - File Upload". Please use this to document how you have responded to the comments, and the adjustments you have made. In

order to expedite the processing of the revised manuscript, please be as specific as possible in your response.

- Data accessibility

If you wish to submit your supporting data or code to Dryad (<http://datadryad.org/>), or modify your current submission to dryad, please use the following link:
<http://datadryad.org/submit?journalID=RSOS&manu=RSOS-181604>

- Competing interests

- Authors' contributions

- Acknowledgements

- Funding statement

on behalf of Professor Steve Brown (Subject Editor)
openscience@royalsociety.org

Comments to Author:

Reviewers' Comments to Author:

Reviewer: 1

Comments to the Author(s)

The manuscript describes transcriptomic changes in mammary glands of sheep affected with clinical mastitis compared to healthy mammary tissues. The study resulted in the typical long list of DEGs, that were subsequently grouped into a meaningful functional categorization. Validation of RNA-seq results was done for a few genes with RT-qPCR and immunohistochemistry. The study was well-conducted and as transcriptome data of clinical mastitis in sheep are yet not available it represents an important addition to literature.

Reviewer: 2

Comments to the Author(s)

This is an interesting manuscript studying the transcriptome signature of clinical mastitis in sheep.

Although the manuscript is well written, the sample size is small and the authors used very mild significant thresholds. Stricter significant thresholds-at least adjusting for multiple testing- should be performed for the Differential Expression analysis. All the analysis related to co-expression, pathway and network analysis etc should be repeated using only the DE results after adjusting for multiple testing using for example FDR correction.

Moreover, the whole discussion should be re-written in a more focused way with in depth comparisons among the present studies and previous studies on the topic. The authors also miss recent publications on the topic for example

<https://bmcgenomics.biomedcentral.com/articles/10.1186/s12864-017-3982-1>

Material and Methods:

The authors talk about natural infection-clinical mastitis without clarifying which was the causing agent. Did they perform microbiological analysis to identify the responsible pathogen? Was the responsible pathogen the same for all three cases?

Line 22: The current reference genome version is *Ovis aries* v3.1. The authors mention that they are using Oar_v4.0 genomic dataset. Could the authors give some more details on these?

Line 36: I assume that the $P < 0.05$ is the uncorrected P value and not the adjusted after FDR correction. I am wondering how many genes remain significantly differentially expressed after adjusting for multiple testing using FDR correction.

A $P < 0.05$ is a very mild threshold and the results could be false positive. This should be amended and the genes which are presented as differentially expressed should be the ones which remain significant after correcting for multiple testing.

Did you find in the two groups any differentially spliced genes?

Author's Response to Decision Letter for (RSOS-181604.R0)

See Appendix A.

RSOS-181604.R1 (Revision)

Review form: Reviewer 2

Is the manuscript scientifically sound in its present form?

Yes

Are the interpretations and conclusions justified by the results?

Yes

Is the language acceptable?

Yes

Is it clear how to access all supporting data?

Yes

Do you have any ethical concerns with this paper?

No

Have you any concerns about statistical analyses in this paper?

No

Recommendation?

Accept with minor revision (please list in comments)

Comments to the Author(s)

The authors have done a good job addressing previous comments and revising the manuscript accordingly.

I am happy with the current version of the manuscript. My only concern is the English writing of the manuscript which needs some further improvement.

Minor comments below:

Line 15 page 2 Introduction: is this indigenous sheep breed meat or dairy?

Line 52 page 2 M&M: change the "is both caused" with "were both caused"

Page 8 line 48 discussion: change "compare with" to "compare to"

Page 8 line 57 discussion: change "in accord" with "in accordance"

Page 9 lines 53-55: the sentence should be re-written to make better sense.

Decision letter (RSOS-181604.R1)

30-May-2019

Dear Dr Li:

On behalf of the Editors, I am pleased to inform you that your Manuscript RSOS-181604.R1 entitled "Digital gene expression analyses of mammary glands from meat ewes naturally infected with clinical mastitis" has been accepted for publication in Royal Society Open Science subject to minor revision in accordance with the referee suggestions. Please find the referees' comments at the end of this email.

The reviewers and Subject Editor have recommended publication, but also suggest some minor revisions to your manuscript. Therefore, I invite you to respond to the comments and revise your manuscript.

- Ethics statement

- Data accessibility

<http://datadryad.org/submit?journalID=RSOS&manu=RSOS-181604.R1>

- Competing interests

- Authors' contributions

- Acknowledgements

- Funding statement

Because the schedule for publication is very tight, it is a condition of publication that you submit the revised version of your manuscript before 08-Jun-2019. Please note that the revision deadline will expire at 00.00am on this date. If you do not think you will be able to meet this date please let me know immediately.

- 1) A text file of the manuscript (tex, txt, rtf, docx or doc), references, tables (including captions) and figure captions. Do not upload a PDF as your "Main Document".
- 2) A separate electronic file of each figure (EPS or print-quality PDF preferred (either format should be produced directly from original creation package), or original software format)
- 3) Included a 100 word media summary of your paper when requested at submission. Please ensure you have entered correct contact details (email, institution and telephone) in your user account
- 4) Included the raw data to support the claims made in your paper. You can either include your data as electronic supplementary material or upload to a repository and include the relevant doi within your manuscript

5) All supplementary materials accompanying an accepted article will be treated as in their final form. Note that the Royal Society will neither edit nor typeset supplementary material and it will be hosted as provided. Please ensure that the supplementary material includes the paper details where possible (authors, article title, journal name).

Kind regards,
Alice Power
Royal Society Open Science
openscience@royalsociety.org

on behalf of Steve Brown (Subject Editor)
openscience@royalsociety.org

Reviewer comments to Author:
Reviewer: 2

Comments to the Author(s)

The authors have done a good job addressing previous comments and revising the manuscript accordingly.

I am happy with the current version of the manuscript. My only concern is the English writing of the manuscript which needs some further improvement.

Minor comments below:

Line 15 page 2 Introduction: is this indigenous sheep breed meat or dairy?
Line 52 page 2 M&M: change the "is both caused" with "were both caused"
Page 8 line 48 discussion: change "compare with" to "compare to"
Page 8 line 57 discussion: change "in accord" with "in accordance"
Page 9 lines 53-55: the sentence should be re-written to make better sense.

Comments from the Editorial Office:

For information about language editing services endorsed by the Royal Society, please follow the link below:

<https://royalsociety.org/journals/authors/language-polishing/>

Author's Response to Decision Letter for (RSOS-181604.R1)

See Appendix B.

Decision letter (RSOS-181604.R2)

04-Jun-2019

Dear Dr Li,

I am pleased to inform you that your manuscript entitled "Digital gene expression analyses of mammary glands from meat ewes naturally infected with clinical mastitis" is now accepted for publication in Royal Society Open Science.

on behalf of Prof Steve Brown (Subject Editor)
openscience@royalsociety.org

Appendix A

Dear Editor Prof. Steve Brown and Reviewers,

On behalf of my co-authors, we thank you very much for your time and effort put in reviewing our manuscript (ID: RSOS-181604.R1) entitled “Digital gene expression analyses of mammary glands from meat ewes naturally infected with clinical mastitis”. We have revised carefully our manuscript after reading the comments and suggestions provided by the two independent reviewers. Those comments have helped us to greatly improve the quality of our manuscript. The revised portion are marked in red in the revised version of this manuscript. The responds to the reviewer's comments and suggestions are as following.

Special thanks again to your reviews. Please do not hesitate to contact us if you have any questions regarding revised version of this paper or responses to reviewer's comments and suggestions.

With best regards,

Taotao Li

Gansu Agricultural University

E-mail: litt@st.gsau.edu.cn

yjma@gsau.edu.cn

Response to Reviewer 2 Comments

Point 1: This is an interesting manuscript studying the transcriptome signature of clinical mastitis in sheep. Although the manuscript is well written, the sample size is small and the authors used very mild significant thresholds. Stricter significant thresholds-at least adjusting for multiple testing- should be performed for the Differential Expression analysis. All the analysis related to co-expression, pathway and network analysis etc should be repeated using only the DE results after adjusting for multiple testing using for example FDR correction.

Response 1: Thank you very much for your comments. Admittedly, in the present study, the sample size is small, but it meets the minimum biological research requirement. In many relevant literatures published recently, the sample size is also n=3. **Reference 1:** Zhang H, Jiang H, Fan Y, Chen Z, Li M, Mao Y, Karrow NA, Loor JJ, Moore S, Yang Z. 2018 Transcriptomics and iTRAQ-proteomics analyses of bovine mammary tissue with *Streptococcus agalactiae*-induced mastitis. *Agric. Food. Chem.* **66**, 11188-11196 (doi: 10.1021/acs.jafc.8b02386). **Reference 2:** Chopradewasthaly R, Korb M, Brunthaler R, Ertl R. 2017 Comprehensive RNA-Seq profiling to evaluate the sheep mammary gland transcriptome in response to experimental *Mycoplasma agalactiae* infection. *PLoS One* **12**, e0170015. (doi: 10.1371/journal.pone.0170015). **Reference 3:** Ma L, Zhang M, Jin Y, Erdenee S, Hu L, Chen H, Cai Y and Lan X. 2018 Comparative transcriptome profiling of mRNA and lncRNA related to tail adipose tissues of sheep. *Front. Genet.* **9**, 365. (doi: 10.3389/fgene.2018.00365). **Reference 4:** Wang XG, Ju ZH, Hou MH, Jiang Q, Yang CH, Zhang Y, Sun Y, Li RL, Wang CF, Zhong JF, Huang JM. 2016 Deciphering transcriptome and complex alternative splicing transcripts in mammary gland tissues from cows naturally infected with *Staphylococcus aureus* mastitis. *PLoS One* **11**, e0167666. (doi: 10.1371/journal.pone.0167666).

In our ongoing study, we selected more sample size (n=5) to reveal the specific regulator mechanisms of key candidate genes and pathways in mastitic sheep mammary glands experimentally infected with single or mixed bacteria isolated from infected animals used in this study.

According to your comment, we re-screen the differentially expressed genes in healthy and mastitic groups based on thresholds of $|\log_2(\text{fold change})| > 1$ and $\text{FDR} < 0.05$. All results related to clustering heatmap, GO annotation, pathway enrichment, co-expression network, protein-protein interaction network etc were also re-analyzed according to differential gene results after FDR correction.

Point 2: The whole discussion should be re-written in a more focused way with in depth comparisons among the present studies and previous studies on the topic. The authors also miss

recent publications on the topic for example
<https://bmcgenomics.biomedcentral.com/articles/10.1186/s12864-017-3982-1>

Response 2: Thank you very much for your comment and suggestion. We have re-written the whole “Discussion” in our revised version according to the comment. In revised manuscript, we also cited this recent reference in “Discussion” part.

Point 3: Material and Methods: The authors talk about natural infection-clinical mastitis without clarifying which was the causing agent. Did they perform microbiological analysis to identify the responsible pathogen? Was the responsible pathogen the same for all three cases?

Response 3: Thank you very much for your comment. We have to apologize for not description in our “Material and Methods”. During experiment, sheep milk samples derived from right udder of 45 clinical cases were collected, and pathogens from each sample were isolated and identified by Gram staining and biochemical methods. In this study, we selected three mastitic sheep were all caused by mixed infections with *Staphylococcus aureus* and *Escherichia coli*.

Point 4: Line 22: The current reference genome version is *Ovis aries* v3.1. The authors mention that they are using Oar_v4.0 genomic dataset. Could the authors give some more details on these?

Response 4: Thank you very much for your comment. We are very sorry for our negligence. In our revised manuscript, we have provided the URL address on Oar_v4.0 genomic dataset used in this study. For details, please see “3.3. Quality control of raw data and mapping reads to the genome” section.

Point 5: Line 36: I assume that the $P < 0.05$ is the uncorrected P value and not the adjusted after FDR correction. I am wondering how many genes remain significantly differentially expressed after adjusting for multiple testing using FDR correction. A $P < 0.05$ is a very mild threshold and the results could be false positive. This should be amended and the genes which are presented as differentially expressed should be the ones which remain significant after correcting for multiple testing.

Response 5: Thank you very much for your invaluable comments and suggestions. As you can see in the answer of Point 3, in our revised manuscript, we re-identified 420 significantly differentially expressed genes based $FDR < 0.05$. GO terms and KEGG pathways were

reanalyzed, and GO terms and pathways enriched significantly by these differentially expressed genes were re-screened with a criterion corrected $p < 0.05$. We have made correction in revised version of this paper, as shown in “3.5. Functional annotation and pathway analysis of DEGs”.

Point 6: Did you find in the two groups any differentially spliced genes?

Response 6: Thank you very much for your comment. We identified some differentially spliced genes in the two groups by rMATS analysis based on a threshold of FDR < 0.05 , and these results have been supplemented in our revised manuscript. For details, please see “4.6. Identification of genes with differential splicing events” and “figure 7”.

Appendix B

Dear Reviewer 2,

Thank you very much for your comments concerning our manuscript entitled “Digital gene expression analyses of mammary glands from meat ewes naturally infected with clinical mastitis”. We have revised this manuscript according to your comments. Additionally, English writing has been improved by a native English speaker in the revised manuscript. The revised portion was marked in red in revised version of this manuscript. The responses to your comments are as following.

Special thanks again to your comments and suggestions.

With best regards,

Taotao Li

Gansu Agricultural University, China

E-mail: litt@st.gsau.edu.cn

Response to Reviewer 2 Comments

Point 1: Line 15 page 2 Introduction: is this indigenous sheep breed meat or dairy?

Response 1: Thank you very much for your comments. We have made correction in the manuscript, as shown in the second sentence from the end paragraph 1 in the “Introduction”.

Point 2: Line 52 page 2 M&M: change the “is both caused” with “were both caused”

Response 2: Thank you very much for your comments. We have made correction in the manuscript, as shown in section “3.1. Experimental animals and design”.

Point 3: Page 8 line 48 discussion: change “compare with” to “compare to”

Response 3: Thank you very much for your comments. We have made correction in the manuscript, as shown in paragraph 4 in the “Discussion”.

Point 4: Page 8 line 57 discussion: change “in accord” with “in accordance”

Response 4: Thank you very much for your comments. We have made correction in the manuscript, as shown in paragraph 5 in the “Discussion”.

Point 5: Page 9 lines 53-55: the sentence should be re-written to make better sense.

Response 5: Thank you very much for your comments. We have re-written this sentence in our revised manuscript, as shown in sentences 1-2, paragraph 6 in the “Discussion”.